# Evaluation of the Impact of the First Wave of COVID-19 and Associated Lockdown Restrictions on Persons with Disabilities in 14 States of India

**DOI:** 10.3390/ijerph191811373

**Published:** 2022-09-09

**Authors:** Shailaja Tetali, Sureshkumar Kamalakannan, Shilpa Sadanand, Melissa Glenda Lewis, Sara Varughese, Annie Hans, G. V. S. Murthy

**Affiliations:** 1South Asia Centre for Disability and Inclusive Development and Research, Indian Institute of Public Health, Hyderabad 500033, Telanagana, India; 2CBM India Trust, Bengaluru 560018, Karnataka, India; 3Handicap International (Humanity Inclusion), Noida 201307, Uttar Pradesh, India

**Keywords:** COVID-19, lockdown restrictions, disabled persons, public health, vulnerable population, India

## Abstract

Background: There is a paucity of data to assess the impact of the COVID-19 pandemic on persons with disabilities (PwDs) in India. About 27.4 million cases were reported as of 27 May 2021. The continuing pandemic in the form of subsequent waves is expected to have negative repercussions for the disabled globally, particularly in India, where access to health, rehabilitation, and social care services is very limited. Therefore, this study aimed to assess the impact of the COVID-19 pandemic and lockdown restrictions on PwDs in India. Objective: To determine the level of disruption due to COVID-19 and the associated countrywide lockdown restrictions on PwD in India during the first wave. Methods: Using a cross-sectional, mixed-methods approach, data were collected from a representative sample of 403 persons with disabilities in 14 states in India during the COVID-19 first wave at two different points in time (Lockdown and post-lockdown phase). Factors associated with the negative impact were examined using the Chi-square test for associations. The paired comparisons between ‘lockdown’ with the ‘post-lockdown’ phase are presented using McNemar’s test and the marginal homogeneity test to compare the proportions. Additionally, a subsample of the participants in the survey was identified to participate in in-depth interviews and focus group discussions to gain in-depth insights on the study question and substantiate the quantitative findings. The framework approach was used to conduct a thematic analysis of the qualitative data. Results: About 60% of the PwDs found it difficult to access emergency medical services during the lockdown, and 4.6% post lockdown (*p* < 0.001). Likewise, 12% found it difficult to access rehabilitation services during the lockdown, and 5% post lockdown (*p* = 0.03). About 76% of respondents were apprehensive of the risk of infection during the lockdown, and this increased to 92% post lockdown (*p* < 0.001). Parents with children were significantly impacted due to lockdown in the areas of Medical (*p* = 0.007), Rehabilitation (*p* = 0.001), and Mental health services (*p* = 0.001). The results from the qualitative study supported these quantitative findings. PWDs felt that the lockdown restrictions had negatively impacted their productivity, social participation, and overall engagement in everyday activities. Access to medicines and rehabilitation services was felt to be extremely difficult and detrimental to the therapeutic benefits that were gained by them during the pre-pandemic time. None of the pandemic mitigation plans and services was specific or inclusive of PWDs. Conclusions: COVID-19 and the associated lockdown restrictions have negatively impacted persons with disabilities during the first wave in India. It is critical to mainstream disability within the agenda for health and development with pragmatic, context-specific strategies and programs in the country.

## 1. Introduction

India reported about 27.4 million cases of COVID-19 as on 27 May 2021. Strict, nation-wide lockdown restrictions were imposed during the first wave of the pandemic in May 2020 [1,2]. There is a paucity of data to quantify the impact of the pandemic on PwD, who are more vulnerable and at higher risk from COVID-19 [3]. Available evidence suggests that PwD often experience serious risks and consequences due to pandemic situations and they are currently facing unethical disadvantages in rationing for critical care and life-saving treatment due to COVID-19 [4,5]. Lack of disability-inclusive response and preparedness for an emergency has exacerbated the existing structural disparities experienced by PwD. Lockdown restrictions meant additional risk to PwD because of disruption to essential services and support [6,7,8]. This is especially so for a country, such as India where access to health, rehabilitation, and social care services is very limited [9,10,11,12]. The situation is further worsened, as disability is both the cause and consequence of poverty [13,14].

According to the WHO survey conducted in 155 countries in May 2020, prevention and treatment services for disabling non-communicable diseases (NCDs) were severely disrupted since the pandemic [15]. Inaccessibility to health and rehabilitation services, functional decline due to reduced physical activity, and disruption of personal and social networks have increased the vulnerabilities faced by PwD into manifolds [16]. India reported 30% fewer acute cardiac emergencies reaching health facilities in rural areas in March 2020 compared to previous years [15]. The underlying health conditions and exclusion from health care services rendered PwD more vulnerable to multiple health conditions including NCDs [16]. According to the rapid assessment survey conducted by the WHO, 94% of the countries surveyed reported that human and financial resources were diverted towards the mitigation of COVID-19 and many public screening programs were reportedly disrupted [15]. PwD have been excluded from even these initiatives related to mitigation and emergency preparedness [17].

During the nationwide lockdown in India, public transport facilities were suspended, which were still not restored in many places by October 2020 [18]. This has further affected access to health care services for PwD, which was already a challenge in India—owing to the inaccessible physical infrastructure, facilities and information [18]. Rehabilitation services in India are already scarce, being mostly located at urban-centric tertiary level facilities. This makes it particularly challenging for rural dwelling PwD with respect to accessibility, availability, and utilization of rehabilitation services [19]. Although telerehabilitation and teleconsultations emerged as a response to support PwD, the evidence for safe, effective and good quality service was lacking [20].

Lockdowns, and the subsequent loss of employment and financial crisis could cause severe socio-economic distress and mental health conditions [21]. Longer quarantines are shown to have a direct correlation with poor mental health outcomes [21]. The pandemic also had a profound impact on children’s well-being, particularly for children with disabilities. The lockdown also led them to further disadvantage, especially in the field of mainstream education [22]. Many PwD faced difficulty receiving pensions [23]. The growing literature on the impact of COVID-19 on PwD clearly describes the neglect, exclusion and vulnerability experienced by them globally [24].

Against this background, we aimed to empirically explore the impact of COVID-19 and the associated countrywide lockdown restrictions on PwD in India during the first wave and generate evidence to inform actions for health emergency preparedness in the future.

### Objective

To investigate the impact of COVID-19 and the associated countrywide lockdown restrictions on PwD in India during the first wave.

## 2. Methods

A cross-sectional, mixed-methods approach was adopted using a quantitative survey as well as qualitative in-depth interviews and focus group discussions. In this paper we report the quantitative findings first, followed by the qualitative findings.

### 2.1. Quantitative Study

Data collection was carried out during the first wave of COVID-19 at two different points in time—first, during the lockdown restrictions (‘lockdown phase’, May 2020) and was repeated after six weeks (‘Post lockdown phase’, July 2020), when the lockdown restrictions were eased partially in each of the study sites.

#### 2.1.1. Participants

For the ‘lockdown phase’ survey, a minimum sample size of 403 was required to estimate the proportion of the impact of the COVID-19 pandemic and lockdown restrictions on five domains (Medical care; Rehabilitation; Mental health; Education and Livelihood and Social participation and empowerment). The level of significance was fixed at 5%, with a relative precision of 10%, an assumed impact of 50% and a non-response rate of 5%. A list containing 0.15 million PwD was obtained from the International Non-Governmental Organization (NGO) CBM, and its partner NGOs, which have been implementing disability programs for many decades in India. A simple random sampling of PwD (or their carers) was undertaken as a sampling strategy. Structured questionnaires were administered to the PwD, if they could answer questions, or their carers if they could not answer on their own. For the ‘Post lockdown phase’ survey, a minimum sample size of 100 participants was approached (25% of the original sample). A stratified random sampling with near to equal distribution of participants across each state was selected.

#### 2.1.2. Data Collection Tool

A structured survey questionnaire was specifically developed by domain experts, with feedback from people implementing disability programs in the field. It had five domains, with several questions under each domain: Medical (15 questions); Rehabilitation (6); Mental health (11); Education and Livelihood (13); Empowerment and Social participation (6). Questions had closed-ended options (access to services and information, medicines, online consultation, postponement of appointment due to lockdown, fears and sources of support, impact on pensions and participation. Two-day training of data collectors was conducted online. The tool was piloted to see if there was any difficulty in the flow and comprehension of the questions. The survey was conducted telephonically by five trained public health post-graduate interns. Informed verbal consent was obtained from each participant accepting to participate in the study. The questionnaire is included as a Appendix A.

#### 2.1.3. Data Analysis

Descriptive statistics were used to summarize the data. We defined the primary outcome ‘level of disruption’ due to COVID-19 as having a ‘Negative impact’, which was defined based on answers to questions in all five domains, i.e., access to medical care and treatment (those answering yes to any three questions), rehabilitation (those answering yes to any one question), mental health (those answering yes to any four questions), education and livelihood (those answering yes to any three questions) and social empowerment and participation (those answering yes to any two questions). Factors associated with the negative impact were examined using the Chi-square test for associations. The paired comparisons between the ‘lockdown’ with ‘post-lockdown’ phases are presented using McNemar’s test and the marginal homogeneity test to compare the proportions.

### 2.2. Qualitative Study

To gain in-depth insights related to the answers provided in the survey, selected participants were involved in either in-depth interviews (IDI) or Focus Group Discussion (FGD) through telephone/online zoom calls. An experienced qualitative researcher conducted the IDIs and FGDs using separate topic guides developed specifically for this.

#### 2.2.1. Participants

PwD, their carers, program managers, administrators, policymakers and government officials were chosen purposively from a network of disability organizations and related civil societies, to maximize the likelihood of their being aware of the issues of disability. A theoretical sampling strategy was used to recruit respondents for interviews. Individuals deemed to be aware of programs or implementation of disability programs were approached for interviews. They were selected based on their work within departments or agencies supporting different disability initiatives and were interviewed to understand their perspectives.

#### 2.2.2. Data Collection

An interview guide was used and the thematic axes around which the interviews revolved were the same as those of the questionnaire IDIs and FGDs, and were continued until saturation of each concept was reached, and further data collection failed to contribute new information [25]. Debriefing meetings were held among the research team members at the end of each interview to ensure data quality and to share emerging findings. Data from digital recorders and any additional notes taken during interviews were transcribed using Microsoft Word. Each transcript was checked for consistency. The transcripts were randomly compared with the recorded digital files for accuracy. Disagreements or issues needing further clarity were resolved through discussions and the triangulation of data sources. The interview guide is included as a Appendix A.

#### 2.2.3. Data Analysis

Qualitative data were analyzed using the Framework approach. The data audio was first transcribed verbatim after familiarization with the audio recording. Manual thematic analysis was performed for interviews based on exploring both predetermined issues of interest and looking for new issues raised by the respondents [26]. Codes were then identified after reading the first few transcripts and then those codes were used as a template for other transcripts and new codes if any were also identified. Themes representing a domain or topic area were listed and coded based on frequency and order of mention. Open coding was conducted and codes were grouped into categories, and themes were identified as stipulated by Graneheim and Lundman (2004) [27]. An in-depth analysis was carried out for each of the themes and subthemes that emerged from the transcripts.

## 3. Results

### 3.1. Quantitative Findings

During the ‘lockdown phase’, 403 respondents with impairment were surveyed from 14 states across India, representing different regions of the country. For the ‘post lockdown phase’, 107 persons were included. Of the 403 respondents, the average age was 28 years with the minimum and maximum ages ranging from 3 to 67 years. Of these 111 (27.7%) were less than or equal to 19 years old. More than half (*n* = 208, 51.6%) had physical impairment, followed by visual (*n* = 65, 16.1%), intellectual (*n* = 44, 10.9%), speech and hearing (*n* = 37, 9.2%), developmental (*n* = 7, 1.7%) and mental health issues (*n* = 5, 1.2%). About half the respondents were married (*n* = 141, 48.6%) and majority of them (89.4%, *n* = 126) had children. More than half (*n* = 184, 63.4%) of the respondents were employed. About 255 (63.3%) received government pensions, with a median pension of INR 700 (US$9) per month (Table 1).

#### 3.1.1. Medical and Rehabilitation Services

The majority of the respondents said that they did not need medical services during the lockdown and after the easing of lockdown. A higher number of respondents (14%) found it difficult to access emergency medical services during lockdown compared to post-lockdown (1.8%). A similar observation was found in accessing rehabilitation services (during lockdown: 12.1%; post-lockdown: 4.6%). No significant difference was observed in accessing other services during and post lockdown (Table 2).

#### 3.1.2. Mental Health

About 75.7% of the respondents were apprehensive of the risk of infection during the lockdown, and this increased to 91.6% post lockdown. Interruption of caregiving and gender-based violence was felt among respondents and was significantly higher in comparison to during post-lockdown (Table 3).

#### 3.1.3. Education and Livelihood

There was no statistically significant difference in education and livelihood, during and post lockdown and in the participation and social empowerment of PwD. During the lockdown phase, 73.3% of the respondents (parents only) felt that on being confined at home the children felt distressed; 70.5% said that schools being shut down affected the child’s learning; 65.9% of the schools are not providing online teaching to children. Among the ones who received online teaching only 9.3% said that the teaching was not in accessible formats. Among four who reported that the online teaching was not in an accessible format, two had speech and hearing impairment, and one each had development and physical impairment. Out of the total 107 respondents in the post lockdown survey, 72.9% hesitated to go to the hospital because of fear of getting COVID, 86% were scared to go out and meet others and 78.1% said they did not fear the lack of companionship.

More than 90% of the participants with either physical, speech, hearing, or visual impairment were impacted by the lockdown in receiving rehabilitation services. Parents with children were significantly impacted due to lockdown in the areas of Medical (*p* = 0.007), Rehabilitation (*p* = 0.001) and Mental health services (*p* = 0.001). Among respondents with children, 86.8% reported having been affected mentally compared to 73.8% who did not have children. Those receiving pensions were impacted in the areas of Rehabilitation, Education, Livelihood and Social Empowerment. The majority of those employed were impacted due to a lockdown in the areas of Medical, Mental health and Social Empowerment. Among those who were employed, 86.4% were impacted due to a lack of regular medical services. Table 4 provides the factors associated with the impact of the lockdown.

### 3.2. Qualitative Findings

We conducted 11 in-depth interviews (IDIs) and four focus group discussions (FGDs) just after the lockdown period. Table 5 depicts the profile of the respondents.

#### 3.2.1. Difficulties in Daily Life and Management

Respondents felt that COVID-19 itself has not caused as much impact as the lockdown, on PWDs’ lives. They felt it will take time for people to follow physical distancing and hand hygiene measures. Lockdown has had a huge negative psychological impact on PwD, especially due to loss of income.

“It was as such a gradual journey for Persons with disability to become self-dependent. Their livelihood got affected very seriously due to lockdown. They began losing hope and motivation which was gained after years of hard work”. (NGO in-charge, Male)

“I ask them to wash the hands but not everyone pays heed. Only few of them wash their hands for 20 s. When I ask them to wear the masks, they don’t listen, saying that nothing happens to us, it’s a disease from another country, when time comes, everyone has to die!” (Caregiver, Female)

#### 3.2.2. Access

Many faced difficulties in accessing basic necessities, such as food, mainly vegetables and pulses, whereas rice was provided by the government. Access to medicines was difficult mainly due to travel restrictions in the region. Most of the information was not available in an accessible format. The reasons why hand washing or maintaining distance was important were not clearly communicated even through informational messages.

“Some persons with disability have not eaten food for 8 days”, (PRI Member and Caregiver, Female)

“Those who are on medication for mental health or serious disorders were completely dependent on free medicines. Their supplies had dried up, but they were not allowed to go the district hospital for refills”.(Program Manager, Male)

#### 3.2.3. Services

Most of the medical care was exclusively reserved for COVID-19 and there was a shortage of ambulances for PwD. The minimum support and assistance required for PwD have not been added to the COVID-19 screening and treatment plan. Most of the rehabilitation services had come to a halt due to the lockdown, similar to education and therapy services in the community.

“Not everyone has motorcycle or a bicycle, there are difficulties in reaching the hospitals”. (ASHA and Caregiver, Female)

“Parents are worried. Their child was happy earlier, going on a wheelchair to school. Since schools are closed, their child cannot interact with other children”. (Program Officer, NGO, Female)

#### 3.2.4. Participation

Participation in leisure and community activities had drastically come down due to travel restrictions as well as fear of infection among people.

“If the Persons with disability was not a government employee, they lost their jobs”. (Program Officer, NGO, Male)

“COVID-19 has broken our confidence, if 5 PwD are sitting in a group, joking and laughing, and somebody coughs or sneezes, automatically, they may not tell on the outside, but they fear about getting corona”. (Govt Official, Male)

#### 3.2.5. Communication

Helplines were set up, however, not many could not use them due to inaccessible formats of communication.

“The biggest challenge was to reach Persons with disability and fulfil their needs. We developed a helpline number with a psychologist for counselling and guidance”. (Govt Official, Female)

#### 3.2.6. Networks

“Our biggest strength is our DPO. The investment that we did over the years to build their capacity has helped us a lot during this time”. (Program Manager, Female)

“They are still doing it, if someone is hungry, somebody will ensure that that person is fed”. (Senior Advisor, NGO, Female)

#### 3.2.7. Compassion and Government Response

“Government is not prepared for the next wave. They should involve Gram sabhas and bring in different agendas including disability, it they are serious about planning for next disaster”. (NGO in-charge, Person with disability)

Many reported that Government guidelines during the COVID-19 outbreak were not inclusive and their need for assistance in terms of travel or COVID-19 screening or treatment was largely ignored.

“If the government considers Persons with disability’s problems as that of its own family’s, then nobody will be unhappy”. (ASHA, PRI Member and Caregiver, Female)

#### 3.2.8. Finances

The economic impact faced by PwD was uniformly reported by all interviewees. Pensions were affected, loss of livelihood mainly among PwD with small businesses and petty shops. PwD had to face hardships not only through the loss of income but the inability to travel to draw money from banks or to obtain essential groceries. Funding cuts were reported by stakeholders working in the NGO sector.

“My child fell ill and I had no money, I save money without my husband’s knowledge, which I had to take out and use. My husband would have earned if there was no lock down and I would never have utilized my savings, but that could not happen because of the situation. Lockdown has affected very badly economically. The whole root cause is the poor economic situation!” (ASHA, PRI Member and Caregiver, Female)

#### 3.2.9. Positive Impact

Many reported that because of the lockdown, PWD are avoiding buying non-essential items. Families are spending more time together and children are with their parents. Reduction in alcohol use and lesser disputes among family members were perceived to be positive impacts of the lockdown.

“Children got to see a different face/personality of their parents” (Program Manager, Ngo, Male)

“Yes, as we are not buying many things from the market, we are saving some money.” (ASHA, PRI Member and Caregiver, Female)

To summarize the main qualitative findings, participant perceptions reiterated that lockdown has had a much harsher impact on persons with disability, than COVID-19 itself. There was little food and rations, specifically due to the poor financial situation of PwD. Pensions were affected, loss of jobs and livelihood from small businesses and petty shops. The inability to travel to withdraw money from banks to obtain essential groceries further complicated matters. There was a significant disruption in accessing medicines, rehab services and essential health services due to travel bans and no travel passes. Some NGOs helped PwD in providing necessities, such as food kits, dry rations as well as psycho-social support, but it was inadequate.

## 4. Discussion

The COVID-19 pandemic and the associated lockdown restrictions during the first wave had a significant negative impact on the lives of PwD in India. This is evident from the difficulties experienced by them in all aspects of their daily life, such as access to information, medical care and rehabilitation, education, livelihood, and social participation.

The effect of the pandemic on medical care and rehabilitation of PwD was profound. This was similar to other aspects of healthcare, such as access to medicines, hospital appointments and surgical procedures. Almost all respondents needed assistance for daily living and were dependent on family members as their carers. Many NGOs provided food kits, dry rations as well as psycho-social support, but the livelihoods of persons with disability were affected, and many of them had to borrow money during the lockdown, with government disability pensions affecting among a third of the respondents. Other studies have also reported inconsistencies in financial support measures and recommendations for livelihood assistance have emerged from other countries [28,29]. A recent UK survey found that 60% of PwD experienced problems accessing food, medicine, and other necessities, similar to what we found [30]. Three-quarters of our respondents stated that children were distressed with school closures and it had affected learning. Similar findings of difficulty in continued accessible education were reported from South America [31].

Psychological reactions to COVID-19 among our respondents ranged from fear, anxiety, panic, hopelessness and depression, to fear of infection. This led to a feeling of stigma, discrimination, and isolation, combined with issues in relationships, abandonment, and violence. Likewise, a recent online survey of PwD in India showed that three-quarters of the respondents were living with anxiety, depression and suicidal thoughts [27]. The lockdown restrictions hindered PwD to engage effectively in their individual, family and social roles when compared to before the pandemic. This is especially important because organized systems were hardly available to meet the specific needs of PwD even before the pandemic. This could be one of the reasons why only 17% of respondents required rehabilitation compared to 85% requiring assistance in activities of daily living. Families have been the only source of support for PwD both before and during the pandemic. This was confirmed by more than half of the respondents who were confident of managing the situation if lockdowns were to be imposed again.

These findings have several implications for existing health, social care and development systems in India, particularly the need to optimize existing resources for the effective implementation of programs and policies for persons with disabilities. All government websites and communication must be disabled-friendly with digital access to the information for PwD, as laid down in the Accessible India Campaign [3]. It is crucial for politicians, policymakers and programmed planners to operationalize inclusion in the agenda for development in all sectors, not just in health or social welfare. Similar to the recent debates on ‘lives versus livelihood’ during the pandemic for PwD, an inclusive response must be ensured to tackle such emergencies in the future. For this to happen, extensive information is needed to provide for the basic and social needs of PwD [3,29]. Advocacy with the governments on these issues is critical: Enabling telerehabilitation and supporting the needs of persons and children with disabilities; Online counseling for the management of stress, fear and anxiety; special financial assistance, subsidies, furlough schemes and clearing obstacles to avail cooperative loans. Best practices from organic farming and dairy schemes for PwD must be incentivized and the income from these initiatives must be promoted widely. The government should ensure that pensions are not negatively impacted in the future. Online education for children in schools must be provided in accessible formats. To avoid the pressure of buying smart phones by parents, education must be provided in formats that are easy, both economic and technology wise. For instance, special educators could prepare individualized lesson plans for children and train parents through phone or a school website podcast.

To our knowledge, this is the first national-level study to evaluate the impact of the COVID-19 lockdown on persons with disabilities on a wide range of stakeholders. The study was conducted across 14 states of India, with a diverse mix of participants including policymakers, program planners and implementers. The study yielded rich data to substantiate the findings from varied perspectives. Primary data collection through telephonic surveys and interviews during the lockdown could be viewed as a strength, given the context, or a limitation, given the approach, door-to-door surveys could have arguably elicited richer details compared to remote interviews. We could not obtain direct inputs from people with hearing impairment and used carers as a proxy. If we had stratified PwD based on the severity of the disability, we could have obtained insights into the extent and difference, if any, in the degree of impact of the pandemic.

## 5. Conclusions

COVID-19 and the associated lockdown restrictions have negatively impacted persons with disabilities during the first wave in India, as shown by our quantitative and qualitative findings. With a diverse mix of participants including policymakers, program planners and implementers from 14 states in India, this study yielded rich data to substantiate the findings from varied perspectives. Although the stratification of participants based on the severity of disability was not possible, the findings reiterated the following: It is critical to mainstream disability within the agenda for health and development with pragmatic, context-specific strategies and programs in the country. The pandemic has reminded us yet again of the urgent need to generate research evidence targeting a disability-inclusive approach for planning preparedness and mitigation of the subsequent COVID-19 waves as well as future health emergencies.

## Figures and Tables

**Table 1 ijerph-19-11373-t001:** Socio-demographic details of the study population.

Variables	Categories	*n* (%)
Age in years, median (IQR)		28 (19, 36.5)
Sex, *n* (%)	Male	243 (60.3%)
	Female	160 (39.7%)
Type of Impairment, *n* (%)	Physical	208 (51.6%)
	Visual	65 (16.1%)
	Intellectual	44 (10.9%)
	Speech and Hearing	37 (9.2%)
	Multiple ^†^	37 (9.2%)
	Developmental	7 (1.7%)
	Mental	5 (1.2%)
Occupation, *n* (%)	Employed	184 (63.4%)
	Student	37 (12.8%)
	Unemployed	69 (23.8%)
Region of India, *n* (%)	Central	121 (30.0%)
	East	84 (20.8%)
	West	70 (17.4%)
	North	59 (14.6%)
	North East	37 (9.2%)
	South	32 (7.9%)
Marital status, *n* (%)	Never married	143 (49.3%)
	Married	141 (48.6%)
	Divorced	3 (1.0%)
	Widowed	3 (1.0%)
Number with children, *n* (%)		126 (89.4%)
Disability pension per month, INR, median (IQR)		700 (500, 1000)

*n* = sample size; IQR: interquartile range; ^†^ Those with more than one type of impairment. Sample size *n* = 403 for all variables except for age (*n* = 401), marital status (*n* = 290), occupation (>19 yrs, *n* = 290), receiving pension (*n* = 255) and respondents with children (*n* = 141).

**Table 2 ijerph-19-11373-t002:** Difficulty in accessing medical and rehabilitation services during and post-lockdown.

Services	Categories	During Lockdown *n* (%)	Post Lockdown *n* (%)	*p* Value
Outpatient clinics	Yes	29 (27.2%)	30 (28.0%)	0.19
	No	21 (19.5%)	9 (8.4%)	
	Did not need	57 (53.2%)	68 (63.5%)	
Emergency medical services	Yes	15 (14.0%)	2 (1.8%)	<0.001 *
	No	12 (11.1%)	5 (4.6%)	
	Did not need	80 (74.7%)	100 (93.4%)	
Medicines	Yes	19 (17.9%)	20 (18.8%)	0.73
	No	29 (27.3%)	24 (22.6%)	
	Did not need	58 (54.7%)	62 (58.5%)	
Rehabilitation services	Yes	13 (12.1%)	5 (4.6%)	0.03 *
	No	13 (12.1%)	15 (14.0%)	
	Did not need	81 (75.6%)	87 (81.2%)	
Regular BP monitoring	Yes	1 (0.9%)	2 (1.8%)	-
	No	3 (2.8%)	0 (0.0%)	
	Did not need	103 (96.2%)	105 (98.1%)	
Regular sugar monitoring	Yes	3 (2.7%)	1 (0.9%)	<0.001 *
	No	4 (3.7%)	2 (1.8%)	
	Did not need	100 (93.5%)	104 (97.2%)	
Surgical procedures	Yes	1 (0.9%)	0 (0.0%)	-
	No	5 (4.7%)	0 (0.0%)	
	Did not need	101 (94.4%)	107 (100.0%)	
Routine medicines	Yes	32 (35.2%)	38 (41.8%)	0.38
	No	16 (17.6%)	8 (8.8%)	
	Did not need	43 (47.3%)	45 (49.5%)	
If online consultation was useful	Yes	8 (7.5%)	7 (6.6%)	0.14
	No	10 (9.3%)	1 (0.9%)	
	Did not need	89 (83.2%)	99 (92.5%)	

Marginal homogeneity test. * Statistically significant at a 5% level of significance.

**Table 3 ijerph-19-11373-t003:** Comparison of respondents’ feelings during lockdown and post lockdown.

Being Bothered by:	Categories	During Lockdown *n* (%)	Post Lockdown *n* (%)	*p* Value
Fear of Infection	Not at all	26 (24.3%)	9 (8.4%)	<0.001 *
	Moderately	56 (52.3%)	57 (53.3%)	
	A lot	25 (23.4%)	41 (38.3%)	
Interruption of support from caregivers	Not at all	78 (86.7%)	80 (88.9%)	<0.001 *
	Moderately	8 (8.9%)	10 (11.1%)	
	A lot	4 (4.4%)	0 (0.0%)	
Gender based violence	Not at all	94 (89.5%)	102 (97.1%)	0.02
	Moderately	7 (6.7%)	3 (2.9%)	
	A lot	4 (3.8%)	0 (0.0%)	
Fear of infecting others	Not at all	35 (32.7%)	35 (32.7%)	0.82
	Moderately	63 (58.9%)	61 (57.0%)	
	A lot	9 (8.4%)	11 (10.3%)	
Fear of dying	Not at all	47 (43.9%)	55 (51.4%)	0.37
	Moderately	56 (52.3%)	46 (43.0%)	
	A lot	4 (3.7%)	6 (5.6%)	
Lack of support	Not at all	54 (50.5%)	52 (48.6%)	0.40
	Moderately	45 (42.1%)	43 (40.2%)	
	A lot	8 (7.5%)	12 (11.2%)	
Loss of income	Not at all	37 (35.9%)	33 (32.0%)	0.32
	Moderately	33 (32.0%)	29 (28.2%)	
	A lot	33 (32.0%)	41 (39.8%)	

Marginal homogeneity test. * Statistically significant at 5% level of significance.

**Table 4 ijerph-19-11373-t004:** Factors associated with the impact of lockdown.

Variables	Categories	Negative IMPACT									
		Medical	*p* Value	Rehabilitation	*p* Value	Mental Health	*p* Value	Education, Livelihood	*p* Value	Social Empowerment	*p* Value
		Yes		Yes		Yes		Yes		Yes	
Age (yr)	<40	323 (81.8%)	0.31 ^F^	351 (88.9%)	0.14 ^F^	342 (86.6%)	0.19 ^F^	341 (86.3%)	0.20 ^F^	190 (52.5%)	0.68 ^F^
	≥40	4 (66.7%)		4 (66.7%)		4 (66.7%)		4 (66.7%)		2 (40.0%)	
Sex	Male	203 (83.5%)	0.17	214 (88.1%)	0.67	207 (85.2%)	0.40	214 (88.1%)	0.16	111 (50.0%)	0.28
	Female	125 (78.1%)		143 (89.4%)		141 (88.1%)		133 (83.1%)		82 (55.8%)	
Type of impairment	Developmental	7 (100.0%)	0.05 ^F^	4 (57.1%)	0.005 ^F^	5 (71.4%)	0.64 ^F^	6 (85.7%)	0.24 ^F^	2 (33.3%)	0.03 ^F^
	Intellectual	28 (63.6%)		32 (72.7%)		35 (79.5%)		33 (75.0%)		22 (62.9%)	
	Mental	4 (80.0%)		4 (80.0%)		5 (100.0%)		5 (100.0%)		0 (0.0%)	
	Multiple	176 (84.6%)		189 (90.9%)		180 (86.5%)		180 (86.5%)		110 (56.1%)	
	Physical	31 (83.8%)		34 (91.9%)		33 (89.2%)		34 (91.9%)		13 (38.2%)	
	Speech and Hearing	54 (83.1%)		60 (92.3%)		57 (87.7%)		59 (90.8%)		26 (42.6%)	
	Visual	28 (75.7%)		34 (91.9%)		33 (89.2%)		30 (81.1%)		20 (60.6%)	
Occupation	Employed	191 (86.4%)	0.003 ^F^	203 (91.9%)	0.05	202 (91.4%)	0.005	188 (85.1%)	0.75	131 (60.9%)	<0.001
	Unemployed	67 (79.8%)		72 (85.7%)		67 (79.8%)		72 (85.7%)		33 (44.6%)	
	Student	66 (70.2%)		78 (83.0%)		76 (80.9%)		83 (88.3%)		27 (35.5%)	
Marital status	Married	122 (85.9%)	0.08	131 (92.3%)	0.08	125 (88.0%)	0.45	126 (88.7%)	0.25	65 (47.8%)	0.21
	Single	204 (78.8%)		224 (86.5%)		221 (85.3%)		219 (84.6%)		126 (54.5%)	
Children	Yes	114 (83.8%)	0.007	123 (90.4%)	0.001	118 (86.8%)	0.001	119 (87.5%)	0.73	58 (45.7%)	0.51
	No	85 (69.7%)		91 (74.6%)		90 (73.8%)		105 (86.1%)		40 (41.2%)	
Pension	No	214 (83.9%)	0.09	231 (90.6%)	0.03 ^F^	228 (89.4%)	0.60	223 (87.5%)	0.04	139 (56.7%)	0.006
	Yes	43 (93.5%)		46 (100.0%)		43 (93.5%)		35 (76.1%)		36 (78.3%)	

^F^: Fishers exact test. Note: Chi square test was perform ed. Frequency with row percentage is reported.

**Table 5 ijerph-19-11373-t005:** Profile of In-depth Interviewees and Focus Group Discussion Respondents.

		IDIs	FGDs	Total
Number of IDIs and FGDs		11	4	15
Total Number of participants		11	16	27
Sex	Male	9	13	22
	Female	2	3	5
Age	20–40 years	3	8	11
	41–60 years	8	8	16
Sector	Government Official	3	2	5
	Program Managers/Advisor/NGO-In charge	5	8	13
	Persons with disabilities	3	-	3
	Carers	-	6	6

## Data Availability

These data will be made available to others after receiving appropriate approval from the institution. Requests should be sent to murthy.gvs@iiphh.org.

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
