# Peer review of "Evaluation of the Impact of the First Wave of COVID-19 and Associated Lockdown Restrictions on Persons with Disabilities in 14 States of India"

_ijerph, 2022, doi:10.3390/ijerph191811373_

Round 1

Reviewer 1 Report

Fist of all congratulations for this so interesting research! It is an innovative issue of discussion and a well written article. However, I suggest some minor improvements in a general check for spelling and expressional errors in the English language. The introduction can be enriched with similar research on people with disabilities and you can expand it a little. You can also improve the methodology chapter by presenting more information as for example the research questions. The aim of the study could also be clearer. It is not clear for example if you conducted 404 interviews. I believe that there must be a separate table for the qualitative survey demographics. For the research tool there are no information if any pilot study was conducted before its use and the validation (statistics) for any tests. Finally, in the conclusions it could be very useful if you link or respond to the research questions of the study. 

Reviewer 2 Report

Review

 The objetive of the study was To determine the level of disruption due to COVID-19 and the associated countrywide lockdown restrictions on PwD in India during the first wave.

The subject matter and the study is relevant, as it looks at a vulnerable population that has been underserved during the COVID-19 pandemic. The sample size seems appropriate for the type of population.

Some recommendations:

At the content level.

- Although the authors describe the situation in India which has its demographic, economic, social and other singularities, it would be interesting if they could cite some similar studies in other countries or regions besides those mentioned (UK, Latin America, Iran).

- In the results, it seems that there are no substantive changes in education, could you elaborate a little bit on that? You say that there are no statistically significant differences between the two periods studied, but you don't mention if in both periods they were negative. This is important to mention since at least 13% of the respondents were students, and 126 participants had young children (assumed to be school-age children, since it says children). There is not much reference to this in either the survey or in the interviews or focus groups. However, in the discussion, the difficulty of adapting online education and the lack of technological adaptations are mentioned.

At the methodological level

- Specify the age range of the respondents. 27.7% of the participants are under 19 years of age, but it is not known what the minimum and maximum age of the participants of the total sample is.

- It is confusing that in the introduction and in the body of the article it is mentioned that there are 404 participants, but in the Method section, 403 are mentioned.

- With regard to the approval of the project by the ethics committee, mention the folio number of this approval, if it has one.

- In the results section there is a typo, it says: and majority of them (89.4%, n=12) had children. According to the data in table 1 the n=126.

- Tables 2 and 3 are a bit confusing to read and interpret.

 - In the qualitative part of the study, it is not mentioned whether the thematic axes around which the interviews revolve are the same as those of the questionnaire or what kind of trigger questions were used. It is suggested to explain this part further.

 - Revise table 5, the total of Number of IDIs & FGDs says 11 and 2 respectively, but does not add up to 15.

 - Revise the references under the format of the journal.

Reviewer 3 Report

The introduction section did not present scholarly arguments related to challenges and disruption of support services  for person with disabilities. It was mainly focusing on issues of 'typically developing' people. Please present your arguments focusing more on the person with disabilities. Additionally, please add more literature to make the arguments precise and clear for your readers.  

Please change the objective of the study, as determining the level was not the focus according to the discussion and conclusion. The study was explorative in nature. 

There are many themes in qualitative findings; please merge sub themes to prepare main themes.  

Please elaborate the conclusion with 'study limitations' and 'future research directions'
